# A comparison of two types of contrast media used in endoscopic retrograde cholangiopancreatography: A retrospective study

Tomoharu Matsuura[1], Yasushi Hamaya[1]*, Shunya Onoue[1], Satoshi Tamura[1], Natsuki Ishida[2], Mihoko Yamade[1], Shinya Tani[1], Moriya Iwaizumi[3], Satoshi Osawa[2], Takahisa Furuta[4], Ken Sugimoto[1]

1 First Department of Medicine, Hamamatsu University School of Medicine, Higashi-ku, Hamamatsu, Japan, 2 Department of Endoscopic and Photodynamic Medicine, Hamamatsu University School of Medicine, Higashi-ku, Hamamatsu, Japan, 3 Department of Laboratory Medicine, Hamamatsu University School of Medicine, Higashi-ku, Hamamatsu, Japan, 4 Center for Clinical Research, Hamamatsu University School of Medicine, Higashi-ku, Hamamatsu, Japan

* yhamaya@hama-med.ac.jp

**Data Availability Statement:** All relevant data are within the manuscript and its Supporting Information files.

## Abstract

### Background

Post-endoscopic retrograde cholangiopancreatography (ERCP) pancreatitis (PEP) is one of the most serious complications of ERCP. Various procedures can reduce the incidence of PEP, such as wire-guided cannulation, prophylactic pancreatic stent placement, and pre-treatment anal insertion of NSAIDs. Recently, iso-osmolar contrast media (IOCM) have been used for ERCP in several hospitals to reduce the risk of PEP in Japan. However, the effect of IOCM is uncertain because few reports have examined IOCM in relation to PEP.

### Aim

This study aimed to investigate the relationship between contrast media used and the incidence of PEP.

### Methods

This retrospective study included all qualifying patients who had undergone ERCP at Hamamatsu University Hospital between January 2012 and January 2020. This study examined whether there was a difference in the onset of PEP between patients administered IOCM and high osmolar contrast medium (HOCM). Propensity score matching was used to analyze patient characteristics and ERCP procedures. Amidotrizoic acid was used as HOCM and iodixanol as IOCM.

### Results

ERCP was performed on 458 patients, and 830 procedures were conducted. After propensity score matching, 162 patients from the amidotrizoic acid group and 162 patients from the

**Funding:** We received no specific funding for this work. The funders had no role in study design, data collection and analysis, decision to publish, or preparation of the manuscript.

**Competing interests:** The authors have declared that no competing interests exist.

iodixanol group were selected. The incidence of PEP was 10.5% (17) in the amidotrizoic acid group and 9.3% (15) in the iodixanol group ($P = 0.71$). Changes in serum amylase levels post- and pre-ERCP were 240.6 ± 573.8 U/L and 142.7 ± 382.1 U/L in the amidotrizoic acid and iodixanol groups, respectively ($P = 0.072$).

## Conclusion

Iodixanol had no prophylactic effect on PEP and clinical outcomes.

## Introduction

Endoscopic retrograde cholangiopancreatography (ERCP) is an advanced endoscopic procedure for the diagnosis and treatment of hepatobiliary pancreatic disease. Although the use of magnetic resonance imaging with magnetic resonance cholangiopancreatography and endoscopic ultrasound is widespread and beneficial for diagnosis, ERCP is required for procedures such as removal of bile duct stones or biliary obstruction. Various complications of ERCP are known, such as pancreatitis, hemorrhage, cholangitis, and perforation [1,2], and, among these, post-ERCP pancreatitis (PEP) is the most serious complication. Acute pancreatitis requires additional treatment and prolonged hospitalization and may leave healthcare providers vulnerable to malpractice claims. According to previous studies, the incidence of PEP is reported to range from 3.8% to 15.1% [3], severe PEP to range between 0.5% and 1%, and the mortality rate to range from 0.11% to 0.7% [4,5]. Some studies have reported that wire-guided cannulation (WGC), administration of non-steroidal anti-inflammatory drugs (NSAIDS), and placement of pancreatic stents reduce the incidence of PEP [6–13]. Cheung et al. have reported that WGC reduced the risk of PEP to 3.2% compared with 8.7% in contrast guided (8.7%) [14]. Otsuka et al. have shown that low-dose rectal diclofenac administration reduced the incidence of PEP to 3.9% in the diclofenac group compared with 18.9% in the control group ($P = 0.017$) [15]. Tsuchiya et al. have reported that placement of pancreatic duct stents may be effective in preventing PEP [16]. As a new approach, endoscopists in Japan have recently begun using iso-osmolar contrast medium (IOCM) for ERCP. These media are expected to have a preventative effect on acute pancreatitis, as IOCM is less irritating to the bile duct and pancreatic duct epithelium compared with high osmolar contrast medium (HOCM). Iodixanol (Visipaque®), a type of IOCM, and amidotrizoic acid (Urografin®), which is an HOCM, are now widely used in Japanese hospitals. There have been no major issues with the use of IOCM noted other than it being more expensive than HOCM (Table 1).

At our hospital, ERCP was performed using amidotrizoic acid as the contrast medium until August 2016. After September 2016, we switched to iodixanol to reduce the incidence of PEP. However, whether PEP can be prevented through iodixanol use remains unclear as only limited number of studies have investigated whether the use of isotonic contrast media prevents PEP [17–19]. Further studies are required on the usefulness of isotonic contrast media for PEP prophylaxis.

**Table 1. A comparison between iodixanol and amidotrizoic acid.**

| Common name | Product name | Ionic properties | Osmolarity (mosm/kgH$_2$O) | pH | Cost (Yen/mL) |
|---|---|---|---|---|---|
| **Iodixanol** | Visipaque® | Non-ionic | 290 | 6.7–7.7 | 84.4 |
| **Amidotrizoic acid** | Urografin® | Ionic | 1570 | 6.0–7.7 | 21.15 |

This study aimed to retrospectively analyze whether iodixanol reduced the incidence of PEP in 840 ERCP procedures performed at Hamamatsu University School of Medicine and investigate the effect of iodixanol on serum amylase levels post-ERCP, as some studies have reported that the post-ERCP examination serum amylase level may be a predictive marker for PEP [20–22].

## Materials and methods

### Study design

We conducted a retrospective investigation of patient background, contrast media used, and PEP incidence rates for ERCP conducted between June 2012 and January 2020 at the Department of Gastroenterology, Hamamatsu University Hospital.

### Exclusion criteria

We excluded postoperative patients who had undergone balloon-assisted endoscopy for ERCP examination, those who could not undergo endoscopy through the duodenum due to malignant gastrointestinal stenosis, and those who had not been evaluated for serum amylase pre- or post-ERCP.

### Assessment of PEP

The criteria proposed by Cotton et al. [23,24] were used for the diagnosis of PEP and the determination of severity. Diagnosis of mild PEP required the following three conditions. First, the patient had new or worsened abdominal pain. Second, the patient had hyperamylasemia (levels of serum amylase at least three times normal) within 24 h post-procedure. Third, the patient required admission or prolongation of planned admission to 2–3 days. Pancreatitis requiring 4–10 days of hospitalization is diagnosed as moderate PEP. Pancreatitis requiring >10 days of hospitalization or percutaneous drainage or surgery is diagnosed as severe PEP. Moreover, pancreatitis that had development of hemorrhagic pancreatitis, phlegmon, pseudocyst, or infection is diagnosed as severe PEP. Patients who had hyperamylasemia but had no abdominal pain were not diagnosed with PEP.

### Patients

ERCP was performed in 458 patients. A total of 98 patients underwent ERCP twice; 32, three times; 10, four times; and 27, five or more times. As a result, 830 ERCP procedures were performed. Amidotrizoic acid was administered in 442 cases and iodixanol in 398 cases. The indications for ERCP are shown in Table 2. Serum amylase level evaluation was performed the day before ERCP and within 24 h post-ERCP. We investigated whether previously reported factors associated with PEP (prior PEP, sex, previous pancreatitis, age, serum bilirubin levels, pancreatic injection, pancreatic sphincterotomy and endoscopic papillary dilation, chronic pancreatitis, pancreatic duct stenting, bile duct stenting, guidewire insertion into the pancreatic duct, or rectally administered NSAIDs) were associated with the incidence of PEP and serum amylase levels [25,26]. The primary cannulation technique was a contrast injection. The number of cannulations was unknown as this information was not included in the inspection report. Additionally, Sphincter of Oddi dysfunction was not evaluated at our hospital due to its infrequency.

**Table 2. Clinical characteristics of patients who underwent ERCP.**

|  | Amidotrizoic acid | Iodixanol | *P*-value |
|---|---|---|---|
| Number of cases | 432 | 398 | 0.475 |
| Age (year), mean (range) ± SD | 70.8 (14–97) ±11.7 | 73.1 (15–98) ±12.9 | 0.0552 |
| Male/Female, n (%) | 272/160(63.0/37.0) | 256/142(64.3/35.7) | 0.718 |
| Serum total bilirubin level (mg/dl) mean (range) ± SD | 2.45 (0.1–29.4) ±3.5 | 3.02(0.1–26.3) ±3.9 | 0.0366 |
| Cases with normal serum bilirubin (%) | 273 (63.2) | 218 (54.8) | 0.0162 |
| Indications for ERCP |  |  |  |
| Bile duct stone | 234 | 197 | 0.331 |
| Stone removal (%) | 147 (33.3) | 127 (31.9) | 0.555 |
| Benign biliary stricture | 2 | 2 | 1.00 |
| Chronic pancreatitis | 6 | 43 | <0.01 |
| IPMN | 27 | 9 | 0.00576 |
| Pancreatic cancer | 62 | 42 | 0.115 |
| Bile duct cancer | 64 | 74 | 0.162 |
| Malignant biliary stricture * | 11 | 11 | 1.00 |
| Pancreaticobiliary maljunction | 24 | 15 | 0.0637 |
| Metallic stent occlusion | 2 | 4 | 0.131 |
| Others | 0 | 1 |  |
| **Prophylactic procedure for PEP** |  |  |  |
| Administration of NSAIDSs (%) | 25 (5.66) | 150 (37.7) | < 0.01 |
| Prophylactic pancreatic duct stent placement (%) | 37 (8.38) | 88 (22.1) | < 0.01 |

* Biliary tract stenosis due to malignant tumors other than biliary pancreatic disease.

ERCP: Endoscopic retrograde cholangiopancreatography; IPMN, intraductal papillary mucinous neoplasm; NSAIDs, non-steroidal anti-inflammatory drugs; PEP, post-endoscopic retrograde cholangiopancreatography pancreatitis; SD, standard deviation.

## Propensity score matching

This study was not a prospective randomized controlled trial; therefore, several selection biases or confounding factors may have been present. Previous studies have shown that some factors are known to be risk factors of PEP, whereas others are considered to prevent PEP. To adjust for differences in patient- and procedure-related covariates, one-to-one propensity score matching was employed using calipers with a width of 0.2 standard deviations. The covariates were all patient- and procedure-related factors found to be associated with the incidence of PEP and serum amylase levels (prior PEP, sex, a past history of pancreatitis, age, serum bilirubin levels, the pre-ERCP serum amylase level, aim of the ERCP, pancreatic injection, endoscopic sphincterotomy, endoscopic papillary balloon dilatation, the presence or absence of bile duct stones and their removal, chronic pancreatitis, pancreatic duct stenting, bile duct stenting, total procedure time, guidewire insertion into the pancreatic duct, or rectally administered NSAIDs) [25,26].

## Statistical analysis

Statistical analyses of the data were performed using SPSS version 24 (IBM, Armonk, NY, United States) and EZR version 1.33 (Saitama Medical Center, Jichi Medical University, Japan) software [27]. Differences between median values were compared using the Mann–Whitney U and Fisher's exact tests. A *P*-value < 0.05 was considered statistically significant.

### Ethical statement

This retrospective study protocol was reviewed and approved by the Ethics Committee of Hamamatsu University School of Medicine (approval number 21–219). This study was conducted in accordance with Good Clinical Practice principles in accordance with the Declaration of Helsinki. The requirement for informed consent was waived due to the retrospective nature of the study and utilization of anonymous data.

## Results

### Patient characteristics and outcomes prior to propensity score matching

Prior to propensity score matching, the amidotrizoic acid group included 432 (272 males, 160 females) patients and the iodixanol group included 398 (256 males, 142 females) patients. In a comparison of patient characteristics and outcomes prior to propensity score matching, the total serum bilirubin levels differed between the two groups ($P = 0.0366$, $t$-test; Table 2), with more patients having normal bilirubin in the iodixanol group than in the amidotrizoic acid group ($P = 0.0162$, Fisher's exact test; Table 2). Indications for ERCP are shown in Table 2. Amidotrizoic acid was used more frequently than iodixanol for patients with intraductal papillary mucinous neoplasm (IPMN) ($P = 0.00576$, Fisher's exact test; Table 2). Iodixanol was used more frequently than amidotrizoic acid in patients with chronic pancreatitis ($P < 0.01$, Fisher's exact test; Table 2). A significant difference was found in the NSAIDs usage rate and in prophylactic pancreatic stent placement between the iodixanol and amidotrizoic acid groups ($P < 0.001$, Table 2). A difference was noted in the frequencies of these prophylactic procedures between the two groups, as the time we started using iodixanol and the time we began prophylactic procedures for many patients were almost concurrent.

The incidence rate of PEP was 8.8% (38 of 432) in the amidotrizoic acid group and 6.3% (25 of 398) in the iodixanol group. There was no difference in PEP incidence between the two groups ($P = 0.191$, Fisher's exact test; Table 3). There were 26 mild cases, 9 moderate cases, and 3 severe cases in the amidotrizoic acid group and 19 mild cases, 6 moderate cases, and 1 severe case in the iodixanol group ($P = 0.797$, Fisher's exact test). Other clinical factors were not associated with the incidence of PEP ($P > 0.05$, $t$-test).

### Serum amylase levels post-ERCP

The mean (range) ± SD serum average amylase levels post-ERCP were 360.2 (10–3941) ± 537.0 U/L in the amidotrizoic acid group and 257.4 (7–1482) ± 448.4 U/L in the iodixanol group ($P = 0.00299$, $t$-test; Fig 1A). The mean (range) ± SD changes in serum amylase levels post- and pre-ERCP were 223.3 (-2225–3836) ± 546.1 U/L in the amidotrizoic acid group and 104.8 (-942–1372) ± 503.8 U/L in the iodixanol group ($P = 0.00126$, $t$-test; Fig 1B). The rate of change in serum amylase levels post- and pre-ERCP was 4.63-fold in the amidotrizoic acid group and 3.01-fold in the iodixanol group ($P = 0.00197$, $t$-test; Fig 1C). Other clinical factors were further examined; however, they were not shown to affect amylase levels post-ERCP ($P > 0.05$, $t$-test).

### Patient characteristics and outcomes after propensity score matching

After propensity score matching, 162 patients were selected for both the amidotrizoic acid and iodixanol groups (Table 3). The duration of hospitalization mean (range) ± SD was 17.5 (1–66) ± 22.2 days in the amidotrizoic acid group and 13.8 (1–256) ± 24.2 days in the iodixanol group; no significant difference was found ($P = 0.151$). The average number of fasting before resuming feeding was 6.04 in the iodixanol/IOCM group and 6.27 in the amidotrizoic acid/

**Table 3. Clinical characteristics of patients after propensity score matching.**

| | Amidotrizoic acid | Iodixanol | *P*-value |
|---|---|---|---|
| Number of cases | 162 | 162 | |
| Age (year), mean (range) ± SD | 70.4(35–92) ±10.46 | 72.0 (32–93) ±12.6 | 0.211 |
| Male / Female, n (%) | 115/47(71.0/29.0) | 108/54(66.7/33.3) | 0.402 |
| Serum total bilirubin level (mg/dl) mean (range) ± SD | 3.05 (0.1–29.4) ±4.59 | 3.00(0.1–26.3) ±4.06 | 0.909 |
| Cases with normal serum bilirubin (%) | 87 (53.7) | 100 (61.7) | 0.216 |
| Indications for ERCP | | | NA |
| Bile duct stone | 61 | 53 | |
| Stone removal (%) | 36(22.2) | 34(21.1) | |
| Benign biliary stricture | 0 | 0 | |
| Chronic pancreatitis | 4 | 13 | |
| IPMN | 10 | 9 | |
| Pancreatic cancer | 33 | 24 | |
| Bile duct cancer | 34 | 41 | |
| Malignant biliary stricture * | 8 | 7 | |
| Pancreaticobiliary maljunction | 12 | 15 | |
| Metallic stent occlusion | 0 | 0 | |
| Others | 0 | 0 | |
| **Prophylactic procedure for PEP** | | | |
| Administration of NSAIDSs (%) | 21 (13.0) | 18 (11.1) | 0.733 |
| Prophylactic pancreatic duct stent placement (%) | 22 (13.6) | 26 (16.0) | 0.535 |
| Number of fasting times mean (range) ± SD | 6.27 (2–42) ± 6.28 | 6.04(1–42) ± 6.98 | 0.757 |

HOCM group (*P* = 0.499). The incidence rate of PEP was 10.5% (17 of 162) in the amidotrizoic acid group and 9.3% (15 of 162) in the iodixanol group. No difference was observed in PEP incidence between the two groups (*P* = 0.714, Fisher's exact test; Table 4).

The mean (range) ± SD serum average amylase levels post-ERCP were 350.6 (10–3941) ± 580.0 U/L in the amidotrizoic acid group and 256.4 (25–3092) ± 375.8 U/L in the iodixanol group (*P* = 0.082, *t*-test; Fig 2A). The mean (range) ± SD changes in the serum amylase levels

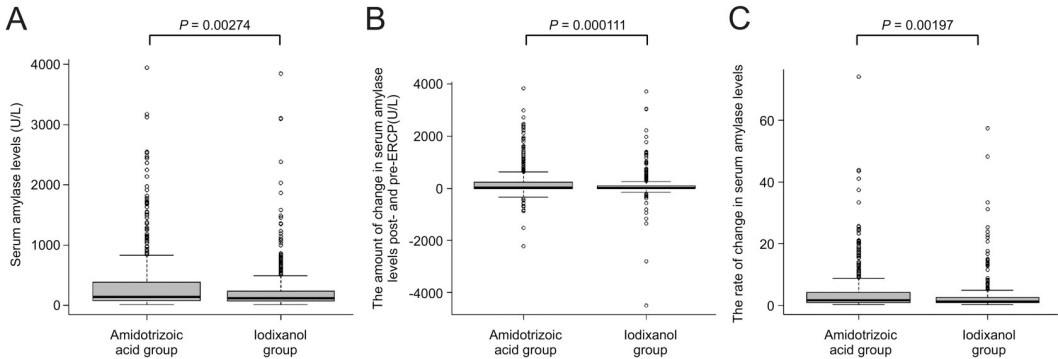

**Fig 1. (A)** Serum amylase levels post-ERCP in the amidotrizoic acid and iodixanol groups prior to propensity score matching. **(B)** The amount of change in serum amylase levels post- and pre-ERCP (post-ERCP amylase levels were subtracted from pre-ERCP amylase levels) in the amidotrizoic acid and iodixanol groups. **(C)** The rate of change in serum amylase levels post- and pre-ERCP (post-ERCP amylase levels were divided by pre-ERCP amylase levels) in the amidotrizoic acid and iodixanol groups.

**Table 4. The incidence of PEP in the amidotrizoic acid and iodixanol groups after propensity score matching.**

| | Amidotrizoic acid | Iodixanol | p-value |
|---|---|---|---|
| Overall PEP (%) | 17 (10.5) | 15 (9.3) | 0.853 |
| Mild PEP (%) | 12 (7.40) | 12 (7.40) | |
| Moderate PEP (%) | 5 (3.09) | 3 (1.85) | |
| Severe PEP (%) | 0 (0) | 0 (0) | |

PEP, post-endoscopic retrograde cholangiopancreatography pancreatitis.

pre- and post-ERCP were 220.3 (-837–3836) ± 565.5 U/L in the amidotrizoic acid group and 142.7 (-1168–3028) ± 382.1 U/L in the iodixanol group (P = 0.072, *t*-test; Fig 2B). The rate of change in serum amylase levels post- and pre-ERCP was 4.00-fold in the amidotrizoic acid group and 3.30-fold in the iodixanol group (*P* = 0.288, *t*-test; Fig 2C). No difference was found between the two groups.

## Discussion

In this study, we retrospectively investigated the effects of different contrast media on the incidence of PEP in 840 procedures. The overall PEP incidence rate was 7.50% (63 of 840), which is similar to rates reported elsewhere [3].The PEP incidence rate in the IOCM group did not differ from that in HOCM group.

PEP is the most serious adverse event of ERCP. It can occur even if ERCP has been performed as planned and without complications or failure, and it can have serious outcomes for patients. Several methods have previously been developed to reduce the risk of PEP, including administration of NSAIDs, prophylactic pancreatic stents, and appropriate cannulation. In addition to these prophylactic methods, the use of IOCM is expected to reduce the incidence of PEP, because IOCM is less irritating to bile and pancreatic duct epithelia compared with HOCM. Several reports have investigated the effects of IOCM on the development of PEP [17–19]. Ogura et al. have conducted a prospective study of 176 patients to investigate the relationship between differences in contrast media and the incidence of PEP and reported no difference in the incidence of pancreatitis [18]; however, severe acute pancreatitis was more frequent in the HOCM group. In contrast, Nagashima et al. have conducted a retrospective

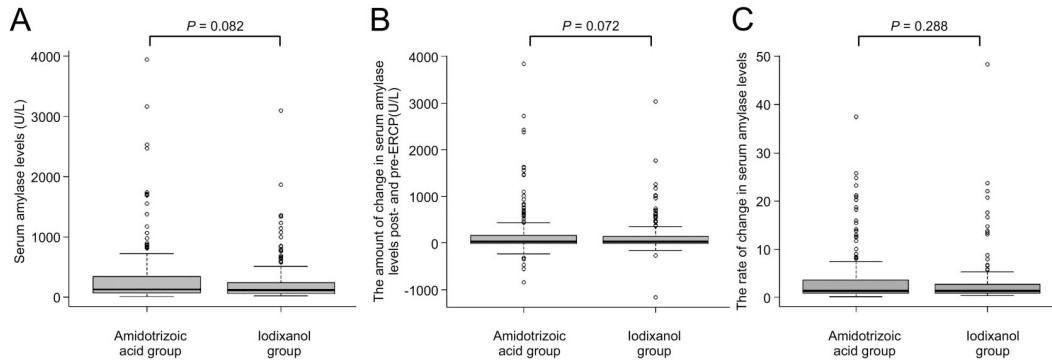

**Fig 2. (A)** Serum amylase levels post-ERCP in the amidotrizoic acid and iodixanol groups after propensity score matching. **(B)** The amount of change in serum amylase levels post- and pre-ERCP (post-ERCP amylase levels were subtracted from pre-ERCP amylase levels) in the amidotrizoic acid and iodixanol groups. **(C)** The rate of change in serum amylase levels post- and pre-ERCP (post-ERCP amylase levels were divided by pre-ERCP amylase levels) in the amidotrizoic acid and iodixanol groups.

study of 291 patients and reported no difference in the incidence of pancreatitis [19]. Furthermore, a meta-analysis by George et al. has shown that there was no difference in the frequency of PEP between low osmotic and high-contrast media groups [17]. Therefore, whether differences in contrast media are associated with the incidence of PEP remains unclear.

As a secondary endpoint, serum amylase levels post-ERCP were compared between the IOCM and HOCM groups. Elevated serum amylase levels are common post-ERCP and occur in up to 75% of patients; however, post-ERCP serum amylase levels have been shown to be associated with PEP. Gottlieb et al. have reported that, at a cutoff value of 276 U/L, sensitivity to PEP was 82%, with a specificity of 76% [20]. Kapetanos have revealed that sensitivity was 72%, with a specificity of 79% when the cutoff value was three times the normal upper limit [22], and Hayashi et al. have reported that sensitivity was 85.5% when the cutoff value was twice the normal upper limit, with a specificity of 85.8% [21]. In clinical practice, extension of the fasting period and administration of protease inhibitors are strategies frequently used for patients with hyperamylasemia post-ERCP, even when they do not meet the diagnostic criteria for PEP. Prevention of hyperamylasemia may reduce the unnecessary treatment of patients. In our study, however, IOCM was not found to have suppressed elevated serum amylase levels post-ERCP.

## Limitations

This study had several limitations. First, it was a single-center retrospective study. Second, ERCP with HOCM and ERCP with IOCM were performed at different times, with ERCP with IOCM more recently performed. The use of NSAIDS and stent placement did not affect serum amylase levels in the IOCM group; however, the development of other ERCP procedures may have influenced our study results. Furthermore, our study was conducted at a teaching hospital; therefore, ERCP was performed by a large number of endoscopists including those who were inexperienced. ERCP performed by inexperienced endoscopists has been reported to be a risk factor for PEP, and the quality of the procedure may have influenced this study [7]. Third, blood tests for serum amylase levels were performed the morning after ERCP had been performed, regardless of the test time. Thus, there was a difference of up to 10 h between the end of the ERCP and blood test. Through reducing the time elapsed between the ERCP and the blood tests, differences between IOCM and HOCM on post-ERCP hyperamylasemia may become clearer.

## Conclusions

Iodixanol did not have a protective effect on PEP. Furthermore, ERCP using iodixanol was not associated with lowering serum amylase elevations. No benefit was found in switching from amidotrizoic acid to iodixanol.

## Supporting information

**S1 File.**
(XLSX)

**S2 File.**
(XLSX)

## Author Contributions

**Conceptualization:** Tomoharu Matsuura, Yasushi Hamaya.

**Data curation:** Tomoharu Matsuura, Yasushi Hamaya, Shunya Onoue.

**Formal analysis:** Tomoharu Matsuura, Yasushi Hamaya.

**Investigation:** Tomoharu Matsuura, Yasushi Hamaya.

**Methodology:** Tomoharu Matsuura, Yasushi Hamaya.

**Project administration:** Yasushi Hamaya, Ken Sugimoto.

**Resources:** Tomoharu Matsuura, Yasushi Hamaya, Shunya Onoue, Satoshi Tamura, Natsuki Ishida, Moriya Iwaizumi, Satoshi Osawa.

**Software:** Yasushi Hamaya.

**Supervision:** Yasushi Hamaya.

**Validation:** Tomoharu Matsuura, Yasushi Hamaya, Shinya Tani.

**Visualization:** Tomoharu Matsuura, Yasushi Hamaya.

**Writing – original draft:** Tomoharu Matsuura, Yasushi Hamaya.

**Writing – review & editing:** Yasushi Hamaya, Shunya Onoue, Satoshi Tamura, Natsuki Ishida, Mihoko Yamade, Shinya Tani, Moriya Iwaizumi, Satoshi Osawa, Takahisa Furuta, Ken Sugimoto.

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
