## [Decision Letter · Decision Letter 0]

1 Sep 2022

PONE-D-22-20342Comparison of two types of contrast media on endoscopic retrograde cholangiopancreatography: a retrospective studyPLOS ONE

Dear Dr. Hamaya,

Thank you for submitting your manuscript to PLOS ONE. After careful consideration, we feel that it has merit but does not fully meet PLOS ONE’s publication criteria as it currently stands. Therefore, we invite you to submit a revised version of the manuscript that addresses the points raised during the review process.

Please note that we have only been able to secure a single reviewer to assess your manuscript. We are issuing a decision on your manuscript at this point to prevent further delays in the evaluation of your manuscript. Please be aware that the editor who handles your revised manuscript might find it necessary to invite additional reviewers to assess this work once the revised manuscript is submitted. However, we will aim to proceed on the basis of this single review if possible. The reviewer has identified some significant concerns regarding the study design and calls for reanalysis of the data to reduce biases. Please respond carefully to these requests, and the other points raised by the reviewer, when preparing your revisions.

We look forward to receiving your revised manuscript.

Kind regards,

Jamie Males

Editorial Office

PLOS ONE

Journal Requirements:

Reviewers' comments:

Reviewer's Responses to Questions

**Comments to the Author**

1. Is the manuscript technically sound, and do the data support the conclusions?

Reviewer #1: Partly

2. Has the statistical analysis been performed appropriately and rigorously? 

Reviewer #1: No

3. Have the authors made all data underlying the findings in their manuscript fully available?

Reviewer #1: Yes

4. Is the manuscript presented in an intelligible fashion and written in standard English?

Reviewer #1: Yes

5. Review Comments to the Author

Reviewer #1: This is a retrospective study of contrast medium for ERCP.

1. As the authors admitted, the study results were highly biased due to its retrospective nature. The authors may want to perform multivariable analysis or propensity score matching to reduce biases.

2. A substantial portion of patients had more than one ERCP in the study cohort. The risk of PEP was low once EST was performed. Please consider a subgroup analysis in cases with native papilla.

3. What is the primary cannulation technique, WGC or contrast injection?

4. Guidewire insertion into the pancreatic duct was a risk factor for PEP (PMID: 34963021, 25442080). Please add this factor in the results and discuss in the context of effect of pancreatic stent for PEP.

5. The cost is high in iodixanol. Please consider cost analysis.

6. Table 3. Prophylactic pancreatic duct stent placement (%) was repeated.

6. PLOS authors have the option to publish the peer review history of their article (what does this mean?). If published, this will include your full peer review and any attached files.

Reviewer #1: No

---

## [Author Response · Author response to Decision Letter 0]

21 Oct 2022

Reviewer #1: This is a retrospective study of contrast medium for ERCP. 1. As the authors admitted, the study results were highly biased due to its retrospective nature. The authors may want to perform multivariable analysis or propensity score matching to reduce biases.

Response: Thank you kindly for your comment and for this valuable advice. Further to your suggestion, we undertook to perform propensity score matching in this study. The manuscript has subsequently been revised as there were no longer differences found in terms of the amylase level and the length of hospital stay. Figure 2 and Table 2 have been revised accordingly.

2. A substantial portion of patients had more than one ERCP in the study cohort. The risk of PEP was low once EST was performed. Please consider a subgroup analysis in cases with native papilla.

Response: Thank you for your kind comment. After propensity matching study, there were 107 of 162 cases with naïve papilla in the amidotrizoic acid group and 112 of 162 cases with naïve papilla in the iodixanol group. In the naïve papilla group, the incidence of PEP was 16 of 107 (15.0%) in the amidotrizoic acid, and 12 of 112 (10.7%) in the iodixanol group (P=0.420). In others, the incidence of PEP was 1 of 55 (1.82%) in the amidotrizoic acid, and 3 of 50 (6.00%) in the iodixanol group (P=0.345). Contrast media did not affect the incidence of PEP.

3. What is the primary cannulation technique, WGC or contrast injection?

Response: Thank you for this important question. Our primary cannulation technique was contrast injection. I have included details concerning this primary cannulation technique in the METHODS section of the manuscript.

4. Guidewire insertion into the pancreatic duct was a risk factor for PEP (PMID: 34963021, 25442080). Please add this factor in the results and discuss in the context of effect of pancreatic stent for PEP.

Response: Thank you for your meaningful comment. After propensity matching study, there were 43 cases in the amidotrizoic acid group with guidewire insertion into the pancreatic duct and 47 cases in the iodixanol group. In the cases with pancreatic stent, the incidence of PEP was 2 of 15 (13.3%) in the amidotrizoic acid, and 7 of 24 (29.2%) in the iodixanol group (P=0.254). In the cases without pancreatic stent, the incidence of PEP was 8 of 28 (28.6%) in the amidotrizoic acid, and 7 of 23 (30.4%) in the iodixanol group (P=1.00). Contrast media was not shown to have affected the incidence of PEP.

As you have pointed out, guidewire insertion into the pancreatic duct and the placement of a pancreatic stent are related to the onset of PEP. However, these results have not been described in this manuscript because we investigated the effect of differences in contrast media on the onset of PEP, and because we adjusted for risk factors using propensity score matching.

5. The cost is high in iodixanol. Please consider cost analysis.

Response: Thank you for your important comment and suggestion. After propensity score matching studies had been undertaken, there was no longer a difference in the length of hospital stay. The average duration of hospital stay was 13.8 days in the iodixanol/IOCM group and 17.5 days in amidotrizoic acid/HOCM group. Iodixanol (Visipaque 100 ml) is 6,000 yen more expensive than amidotrizoic acid (Urografin 100 ml). In Japan, hospitalization costs approximately 8,000 yen per day, but there was no difference in cost because there was no difference in hospital stay (P = 0.44).

6. Table 3. Prophylactic pancreatic duct stent placement (%) was repeated.

Response: Thank you for pointing this out. The duplicate information has been deleted, as has our original Table 3, with the information contained now added to Table 2.

---

## [Decision Letter · Decision Letter 1]

8 Nov 2022

PONE-D-22-20342R1A comparison of two types of contrast media used in endoscopic retrograde cholangiopancreatography: a retrospective studyPLOS ONE

Dear Dr. Hamaya,

Thank you for submitting your manuscript to PLOS ONE. After careful consideration, we feel that it has merit but does not fully meet PLOS ONE’s publication criteria as it currently stands. Therefore, we invite you to submit a revised version of the manuscript that addresses the points raised during the review process.

We look forward to receiving your revised manuscript.

Kind regards,

Kenji Fujiwara, Ph.D., M.D.

Academic Editor

PLOS ONE

Additional Editor Comments

Dear Dr. Hamaya.

The article is a study about contrast media of ERCP in order to reduce common and severe complication, ERCP pancreatitis (PEP). The authors compared two types of contrast media as a retrospective study. The article is well-written and the authors responded appropriately to the questions from one reviewer. I reviewed the article as the editor and also one reviewer. I have some questions and I think the manuscript needs some revision. I summarized my opinions.

1. The article did not include the details of the criteria for diagnosis of PEP. The information of the reference was shown, but I recommend the authors should show an easy summary of the criteria. Especially, the level of serum amylase is not always necessary for the diagnosis of PEP. For easy understanding of readers, the authors should make clear the difference between PEP and high serum amylase levels in the manuscript. In addition, the authors should compare the other factors of diagnosis of PEP when the authors compared the serum factor which is just one factor of the diagnosis.

2. 13.8-17.5 days of hospital stays sound long. I guess the hospital stays may include other treatments like cholecystectomy or rehabilitation. If so, I am not sure the comparison of hospital stays between the two groups has meaning. If the authors wanted to compare the complications of ERCP by focusing on two contrast media, they may use other factors, like the period of high serum amylase level.

Reviewers' comments:

Reviewer's Responses to Questions

**Comments to the Author**

1. If the authors have adequately addressed your comments raised in a previous round of review and you feel that this manuscript is now acceptable for publication, you may indicate that here to bypass the “Comments to the Author” section, enter your conflict of interest statement in the “Confidential to Editor” section, and submit your "Accept" recommendation.

Reviewer #1: All comments have been addressed

2. Is the manuscript technically sound, and do the data support the conclusions?

Reviewer #1: Yes

3. Has the statistical analysis been performed appropriately and rigorously? 

Reviewer #1: Yes

4. Have the authors made all data underlying the findings in their manuscript fully available?

Reviewer #1: Yes

5. Is the manuscript presented in an intelligible fashion and written in standard English?

Reviewer #1: Yes

6. Review Comments to the Author

Reviewer #1: 1. The authors provided a table for patient charcateristics in all cohort alone but please add a table for patient characteristics after propensity matching.

7. PLOS authors have the option to publish the peer review history of their article (what does this mean?). If published, this will include your full peer review and any attached files.

Reviewer #1: No

---

## [Author Response · Author response to Decision Letter 1]

21 Dec 2022

Reviewer #1The article is a study about contrast media of ERCP in order to reduce common and severe complication, ERCP pancreatitis (PEP). The authors compared two types of contrast media as a retrospective study. The article is well-written and the authors responded appropriately to the questions from one reviewer. I reviewed the article as the editor and also one reviewer. I have some questions and I think the manuscript needs some revision. I summarized my opinions.

1. The article did not include the details of the criteria for diagnosis of PEP. The information of the reference was shown, but I recommend the authors should show an easy summary of the criteria. Especially, the level of serum amylase is not always necessary for the diagnosis of PEP. For easy understanding of readers, the authors should make clear the difference between PEP and high serum amylase levels in the manuscript. In addition, the authors should compare the other factors of diagnosis of PEP when the authors compared the serum factor which is just one factor of the diagnosis.

Response: Thank you for your comment and the valuable advice. Further to your suggestion, we have added details of the diagnostic criteria for PEP to the “Materials and Methods” section. PEP was diagnosed in patients with all three of the following criteria: abdominal pain, hyperamylasemia (≥3 times the upper limit of normal), and prolonged hospital stay (required fasting for eight or more meals), according to the cited literature criteria. In the amidotrizoic acid group, 10.5% of the patients (17 of 162) had abdominal pain, and in the iodixanol group, 10.5% of the patients (17 of 162) had abdominal pain (P = 1.00). Further, 27.1% of the patients (44 of 162) required fasting for eight or more meals in the amidotrizoic acid group, and 17.9% of the patients (29 of 162) required fasting for 8 or more meals in iodixanol group (P = 0.062).

2. 13.8-17.5 days of hospital stays sound long. I guess the hospital stays may include other treatments like cholecystectomy or rehabilitation. If so, I am not sure the comparison of hospital stays between the two groups has meaning. If the authors wanted to compare the complications of ERCP by focusing on two contrast media, they may use other factors, like the period of high serum amylase level.

Response: Thank you for your critical comment and suggestion. As noted by the reviewers, the length of hospital stay is strongly influenced by treatment of comorbidities and other diseases. We could not compare the duration of high amylase levels because the timing of blood tests after endoscopy was different in each case. However, we examined the period during which the subjects were able to start eating after ERCP (number of times of fasting after ERCP). The type of contrast media did not affect the duration of fasting (P = 0.499). We have added the result to the “Results.” sections.

6. Review Comments to the Author

Please use the space provided to explain your answers to the questions above. You may also include additional comments for the author, including concerns about dual publication, research ethics, or publication ethics. (Please upload your review as an attachment if it exceeds 20,000 characters) Reviewer #1: 1. The authors provided a table for patient charcateristics in all cohort alone but please add a table for patient characteristics after propensity matching.

Response: Thank you for your comment. We have added new tables about patient characteristics after propensity matching (Table 3).

---

## [Editor Report · Decision Letter 2]

26 Dec 2022

A comparison of two types of contrast media used in endoscopic retrograde cholangiopancreatography: a retrospective study

PONE-D-22-20342R2

Dear Dr. Hamaya,

We’re pleased to inform you that your manuscript has been judged scientifically suitable for publication and will be formally accepted for publication once it meets all outstanding technical requirements.

Kind regards,

Kenji Fujiwara, Ph.D., M.D.

Academic Editor

PLOS ONE

Additional Editor Comments (optional):

Dear Dr. Yousuke Nakai.

Thank you for your prompt re-submission. I think the manuscript is appropriately revised and updated and is eligible for acceptance.

Yours sincerely,

Kenji Fujiwara

Academic editor
---

## [Editor Report · Acceptance letter]

29 Dec 2022

PONE-D-22-20342R2 

A comparison of two types of contrast media used in endoscopic retrograde cholangiopancreatography: a retrospective study 

Dear Dr. Hamaya:

I'm pleased to inform you that your manuscript has been deemed suitable for publication in PLOS ONE. Congratulations! Your manuscript is now with our production department. 

Kind regards, 

on behalf of

Dr. Kenji Fujiwara 

Academic Editor

PLOS ONE